# The Role of Cone-Beam Computed Tomography CT Extremity Arthrography in the Preoperative Assessment of Osteoarthritis

Marion Hamard , Marta Sans Merce, Karel Gorican, Pierre-Alexandre Poletti, Angeliki Neroladaki and Sana Boudabbous *

Division of Radiology, Department of Diagnosis, Geneva University Hospitals, Gabrielle-Perret-Gentil 4, 1205 Geneva, Switzerland; marion.hamard@hcuge.ch (M.H.); marta.sansmerce@hcuge.ch (M.S.M.); karel.gorican@hcuge.ch (K.G.); pierre-alexandre.poletti@hcuge.ch (P.-A.P.); angeliki.neroladaki@hcuge.ch (A.N.)
* Correspondence: sana.boudabbous@hcuge.ch

**Abstract:** Osteoarthritis (OA) is a prevalent disease and the leading cause of pain, disability, and quality of life deterioration. Our study sought to evaluate the image quality and dose of cone-beam computed tomography arthrography (CBCT-A) and compare them to digital radiography (DR) for OA diagnoses. Overall, 32 cases of CBCT-A and DR with OA met the inclusion criteria and were prospectively analyzed. The Kellgren and Lawrence classification (KLC) stage, sclerosis, osteophytes, erosions, and mean joint width (MJW) were compared between CBCT-A and DR. Image quality was excellent in all CBCT-A cases, with excellent inter-observer agreement. OA under-classification was noticed with DR for MJW ($p = 0.02$), osteophyte detection (<0.0001), and KLC ($p < 0.0001$). The Hounsfield Unit (HU) values obtained for the cone-beam computed tomography CBCT did not correspond to the values for multi-detector computed tomography (MDCT), with a greater mean deviation obtained with the MDCT HU for Modeled Based Iterative Reconstruction 1st (MBIR1) than for the 2nd generation (MBIR2). CBCT-A has been found to be more reliable for OA diagnosis than DR as revealed by our results using a three-point rating scale for the qualitative image analysis, with higher quality and an acceptable dose. Moreover, the use of this imaging technique permits the preoperative assessment of extremities in an OA diagnosis, with the upright position and bone microarchitecture analysis being two other advantages of CBCT-A.

**Keywords:** cone-beam computed tomography arthrography; X-ray; osteoarthritis; Kellgren and Lawrence classification; density; radiation





## 1. Introduction

Osteoarthritis (OA) is a prevalent, age-related worldwide disease and the leading cause of pain, disability, and deterioration of quality of life. OA is usually defined based on imaging by means of five hallmarks: joint space narrowing or mean joint width (MJW), subchondral sclerosis, marginal osteophytes, subchondral cysts (geodes), and altered shape of the joint surfaces [1]. Digital radiography (DR) remains the gold standard imaging modality for both the initial evaluation of OA and longitudinal follow-up in clinical practice and research [2–7]. It is more accessible, the least expensive, and the most commonly deployed imaging modality [5] and allows for risk stratification [2]. Radiographic outcome measurements are still the only clinical trial end points approved by regulatory authorities [7]. DR can detect marginal osteophytes, subchondral sclerosis, and cysts and determine the MJW [3–5,7], and it is considered an established determinant of OA severity and longitudinal worsening [2]. However, it is well known that DR (uniplanar modality) is unable to directly visualize OA-associated damage in articular and periarticular non-osseous joint structures, e.g., the meniscus [2,4,5]. The smallest detectable difference of least 0.2 mm average for the MJW of OA knee joints is observed [2].

The most widely and reliably employed semi-quantitative DR assessment for OA is the Kellgren and Lawrence classification (KLC), proposed in 1957 [8]. This method was

initially employed to classify knee OA severity according to five grades (Table 1). As of now, this classification is widely used for all extremity OA grading: OA diagnosis is established if the score is ≥2.3 The most widely employed quantitative DR assessment is the measurement of the joint MJW obtained from knee DR. These assessments have proven reliable, in particular when they extend over a 2-year period and the radiographs are obtained with the knee in a standardized flexed position [7].

**Table 1.** Kellegren and Lawrence classification system for osteoarthritis, as applied in this study.

| Grades | Description |
|---|---|
| 0 | Normal, with no radiographic findings of OA |
| 1 | Doubtful joint space narrowing<br>Doubtful osteophytes |
| 2 | Possible joint space narrowing<br>Definite osteophytes |
| 3 | Multiple and moderately sized osteophytes<br>Definite joint space narrowing<br>Small pseudocystic with sclerotic walls<br>Possible deformity of the bone contour |
| 4 | Multiple and large osteophytes<br>Severe joint space narrowing<br>Marked sclerosis<br>Definite bone deformity |

OA: osteoarthritis.

Multi-detector computed tomography (MDCT) constitutes an imaging modality with several advantages, including excellent analyses of the cortical bone, soft tissue calcifications, and facet joint OA [4,5]. Furthermore, the subchondral trabecular bone architecture can be analyzed, and bone density with calcium crystal deposits can be measured. The method's limitations are the radiation exposure and limited soft tissue evaluation compared with magnetic resonance imaging [5]. MDCT arthrography is the most accurate method for indirectly evaluating superficial and focal cartilage damage and other intrinsic joint structures, especially the central osteophytes that signal more severe OA changes than marginal ones alone. This method displays a high spatial resolution and high contrast among cartilage, superficial, and deep boundaries [4,5]. To resume, 3D imaging is more confident to detect earlier and precise areas of joint loosening. Moreover, for this pre-surgical population, as you know, the technique of replacement depends on the involved zones and the degree of cartilage loss, and CBCT directly visualizes the cartilaginous surfaces. In addition, and inherent to the technique, structures are superposed, for example, between small bones as in the carpus to detect joint narrowing mainly in the sagittal plane.

Cone-beam computed tomography (CBCT), initially applied for dental imaging, has recently emerged as a new dedicated extremity-imaging method [9–11]. CBCT is an emerged technique with a high cost and dose effectiveness in various pathologies; for example, for the temporo-mandibular joint, compared to magnetic resonance imaging (MRI), even this technique remains mandatory for soft tissue diseases [12]. CBCT uses a pyramid-shaped DR beam and flat panel detector that rotates 216.5° around the patient. The main advantages of CBCT are its high spatial resolution [13], which permits a detailed analysis of the bone architecture, lower radiation exposure, and smaller field of view (FOV) compared to MDCT [3,14]. It has been demonstrated that knee joints can now be imaged during both weight-bearing and non-weight-bearing modes, with excellent image quality for bone and good/adequate quality for soft tissues [3,15,16]. The weight-bearing (WB) mode can detect subtle evidence of joint instability [17]. We can easily assume that trabecular and cortical bone qualitative analyses are superior using CBCT than DR and that the superficial and deep cartilage analyses are preferable with CBCT arthrography (CBCT-A). In addition, the main concern of the CBCT technique is the alteration of the

HU with the first generation of the iterative reconstruction compared with MDCT that is inherent to this modality, as further explained in the Section 4; therefore, CBCT apparently cannot be used for estimating bone density [18]. Of note, CBCT-A is considered an invasive method. Indeed, as a prior arthrography needs to be performed, with positive contrast medium (CM) infiltration, potential infectious and hemorrhagic complications may occur.

This study sought to better define CBCT-A's place in OA diagnosis as compared to DR, and to evaluate its clinical image quality, radiation dose, and bone density quantification for the improved use of this technique in the musculoskeletal field. Our hypothesis is that the CBCT-A is superior to DR and an alternative to CT for surgical planning for osteoarthritis.

## 2. Materials and Methods

### 2.1. Patient Population

This prospective study was performed after cantonal ethics committee research (CCER) approval and in accordance with the guidelines of the Helsinki declaration. The number of the CCER-HUG Geneva approval is 2017-01276.

All patients were referred to our institution for CBCT (traumatology, CBCT-A, etc.) from a specialized orthopedic department. They were consecutively and prospectively included during seven months, during which the reconstruction station for second-generation model-based iterative reconstruction (MBIR) was available [19]. Those patients were addressed to assess the pathological joint space in prevision of surgical intervention.

All patients provided informed consent before their participation of this study, all were aged 18 years or more. Only patients who declined consent were excluded from this prospective study.

Overall, 32 patients (men: 23; women: 9; mean age: 75.14 years) were included in the study, all referred from the orthopedic department: 18 wrist cases (two cases with both sides), 9 ankle, and 5 knees for analysis. Two male patients underwent a CBCT-A of each wrist. All patients underwent a CBCT-A and DR of their extremity. All patient data were anonymized.

Indications for the CBCT-A were assessments of OA secondary to the following (Figure 1): osteochondral lesion (OCL) of the talus ($n$ = 4), osteochondritis dissecans of the talus ($n$ = 1), talocrural OA follow-up ($n$ = 1), ligamentous syndesmosis injury ($n$ = 1), after talocrural prosthesis ($n$ = 1), after debridement of OCL of the talus ($n$ = 1), radio-scaphoid OA after wrist traumatism ($n$ = 7) or scaphoid fracture ($n$ = 2), after arthrodesis of the carpal joint ($n$ = 1), scaphoid non-union advanced collapse ($n$ = 7), chronic instability of the triangular fibrocartilage complex ($n$ = 1), knee traumatism ($n$ = 2), tibial fracture ($n$ = 1), tibial OA ($n$ = 1), and after cruciate ligament repair ($n$ = 1).

In our institution, when it was mandatory to assess bone quality, as well as the degree of cartilage damage, CBCT-A was preferred over MRI due the high resolution of the bone and cartilage study. MRI is generally performed to assess ligamentous stability and tendinous injuries, and this category of patients was not included in this study.

Planed surgeries were styloidectomy versus staphyloidectomy or wrist arthrodesis according, respectively, to radio-scaphoid cartilage damage and medio carpal cartilage damage. For the knee, one compartment or total arthroplasty was discussed regarding cartilage preservation. Finally, and for ankle joints, a combination of the upright position and optimal analysis joint allowed for the surgeon to choose between arthroplasty and arthrodesis.

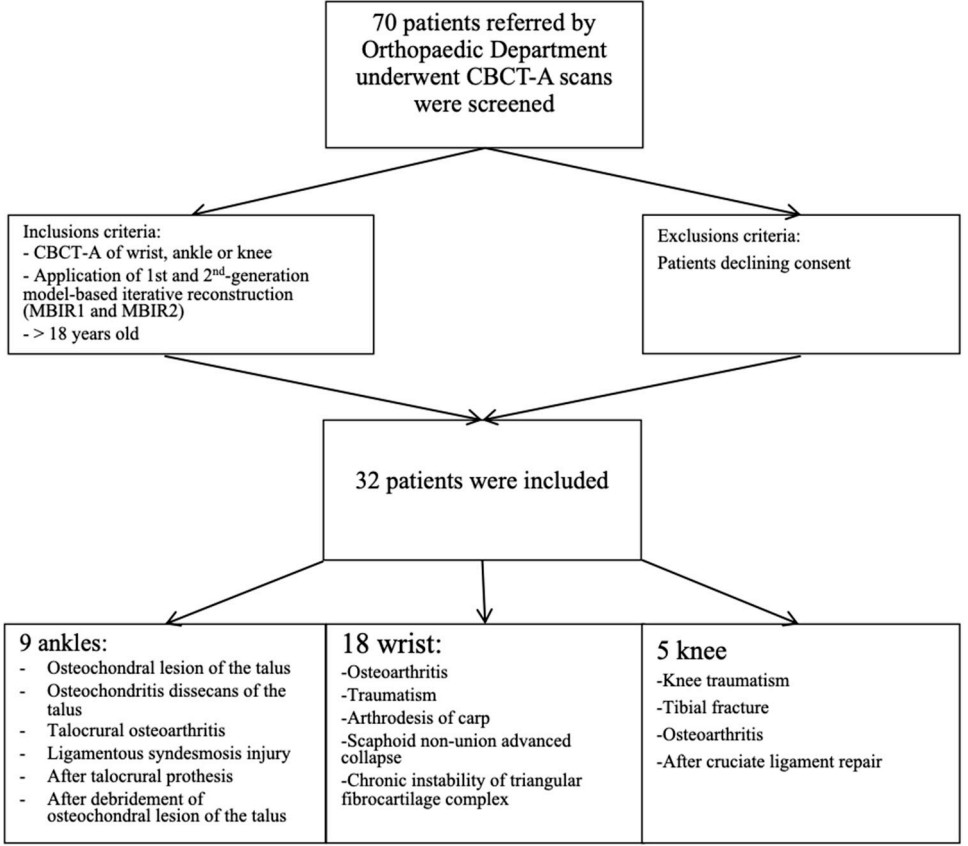

**Figure 1.** Diagram of patient recruitment with inclusion criteria.

## 2.2. DR and CBCT-A Acquisition Protocol and Image Reconstruction

An arthrography was performed 15–20 min before CBCT-A acquisition. For wrist CBCT-A, we performed first percutaneous arthrography in one dedicated fluoroscopic room 20 min before CBCT acquisition. The patient was lying in the "superman" position, with the supine ventral position on the fluoroscopic table. For small joints, we used a 22 G needle for the articular injection of CM. We performed a mix in the syringe of 2 cc of Rapidocaine local anesthesia 1% (Sinetica SA, 200 mg/20 mL, Mendrisio Switzerland) and 8cc of Iopamiron 300 mg/mL (Iopaidolum 3 g iodine/10 mL, Bracco Suisse SA, Geneva, Switzerland). Then, we put the needle in three different localizations under fluoroscopic image-guidance, first in the medio-carpal joint, second in the radio-ulnar joint, and finally in the radio-carpal joint. For ankle joint arthrography, we injected the same mix, but the patient was lying in the dorsal position on the fluoroscopic table, with leg extension, the foot on the table (plantar flexion), and an ankle internal rotation of 20°. For the knee, the patient was lying in the supine position, with a flexion of 30°. The lateral approach was used to inject the CM.

In our department, the CBCT (OnSight, Carestream Health, Rochester, New York, NY, USA) has a gantry featuring a 58 cm patient aperture and movable table enabling an upright position, enabling us to perform a WB CBCT for the lower extremities. The ankle and knee were scanned in the upright position (Figure 2A). The wrist was scanned with the patient sitting and the arm extended (Figure 2B). The acquisition parameters for the wrist are summarized in Table 2. Prior to the acquisition, two scouts were performed, one antero-posterior (AP) and one lateral (LAT). For more simplicity, all CBCT-A images were reconstructed in coronal and sagittal planes with the bone kernel (window width: 1500; window level: 300). All images were first reconstructed with the MBIR1 for routine practice and then reconstructed with MBIR2 in a dedicated research workstation prior to analysis [15]. All images were digitally stored in the PACS and anonymized.

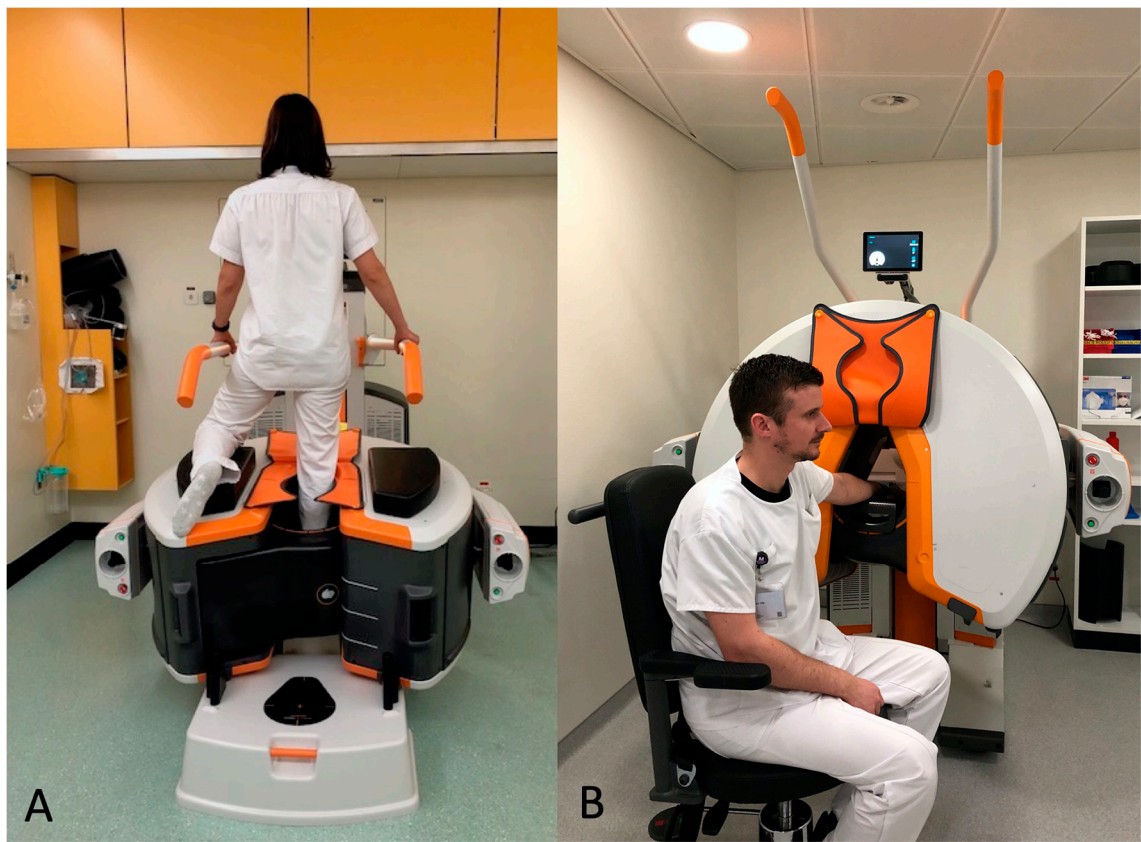

**Figure 2.** Upright position in the WB CBCT to investigate the foot, ankle, or knee, (**A**) and sitting position with arm extended to explore the hand and wrist (**B**). WB: weight-bearing; CBCT: cone-beam computed tomography.

**Table 2.** Scanning parameters of CBCT (OnSight, Carestream Health, Rochester, New York, NY, USA) and DR (Siemens, ISIO and Philips, DigitalDiagnost, Siemens Healthineers International SA, Germany and Philips Netherlands).

| Parameters | CBCT | DR |
|---|---|---|
| Energy | 80 kVp | 50 kVp |
| Current | 5 mA | 3.2 mAs |
| FOV | 216 × 216 mm | |
| Matrix | 884 × 884 | |
| Isotropic voxel size | 0.26 mm | |
| Rotation time | 25.18 s | |
| Exposure time | 21 s approximately | |
| Scan rotation angle | 216.5° | |
| CTDI (indicated) | 3.14 mGy (16 cm phantom) | |
| Focus-detector distance | | 120 cm |

CBCT: cone-beam computed tomography; DR: digital radiography; s: second; CTDI: computed tomography dose index; kVp: kilovoltage peak; mGy: milli-Gray.

For the radiographic technology system, we used DR. The parameters applied for DR are summarized in Table 2. For the wrist, knee, and ankle, two projections were performed, one AP and one LAT, in supine position for the knee and ankle.

*2.3. Qualitative Image Analysis for CBCT-A*

All DR and CBCT-A images were analyzed by two blinded independent MSK radiologists, one with 2 years of experience in MSK imaging and the other one with 5 years. The images were read in a dark reading room using 3D visualization software (Osirix, Rosset,

Geneva, Switzerland, Osirix MD v 13.0.1). All CBCT-A images were analyzed for their overall quality using a three-point rating scale (2: excellent; 1: good; 0: poor).

The KLC stage was compared between the CBCT-A and DR using the usual grading score (Table 1). Erosion and sclerosis were evaluated using a three-point rating scale (0: absence; 1: density change in cortical bone; 3: density change in trabecular bone). Cartilage abnormalities were similarly evaluated using a three-point rating scale (0: normal; 1: thinning; 2: exposed subchondral bone). We used this KLC score for its simplicity. Indeed, the revised Altman atlas is more complete and more objective, but more complicated to be applied in clinical routine practice [16].

### 2.4. Quantitative Image Analysis of CBCT-A

The correspondence between electronic densities and the HU was evaluated using a standardized CIRS-062MA (Norkfold, VA, USA) phantom (Figure 3) for the CBCT images reconstructed with the two available reconstruction algorithms: MBIR 1 and MBIR2. The results were compared to the HU obtained from the two MDCT (SOMATOM Definition Flash Siemens Healthcare and Discovery 750 HD GE Healthcare) available in our department. The phantom comprises nine inserts with different materials corresponding to different electronic densities (Figure 3A). The reproducible region of interest (ROI) (size and seat) was placed in all nine inserts (Figure 3B), with three measurements for every item. Then, we analyzed the HU for all 32 CBCT-As in the coronal plane for the CM, trabecular subchondral bone, cortical bone, and cartilage densities after application of the MBIR2. ROIs were also placed in all cases, with three measurements for every item (Figure 4A). The mean was reported for statistical analysis.

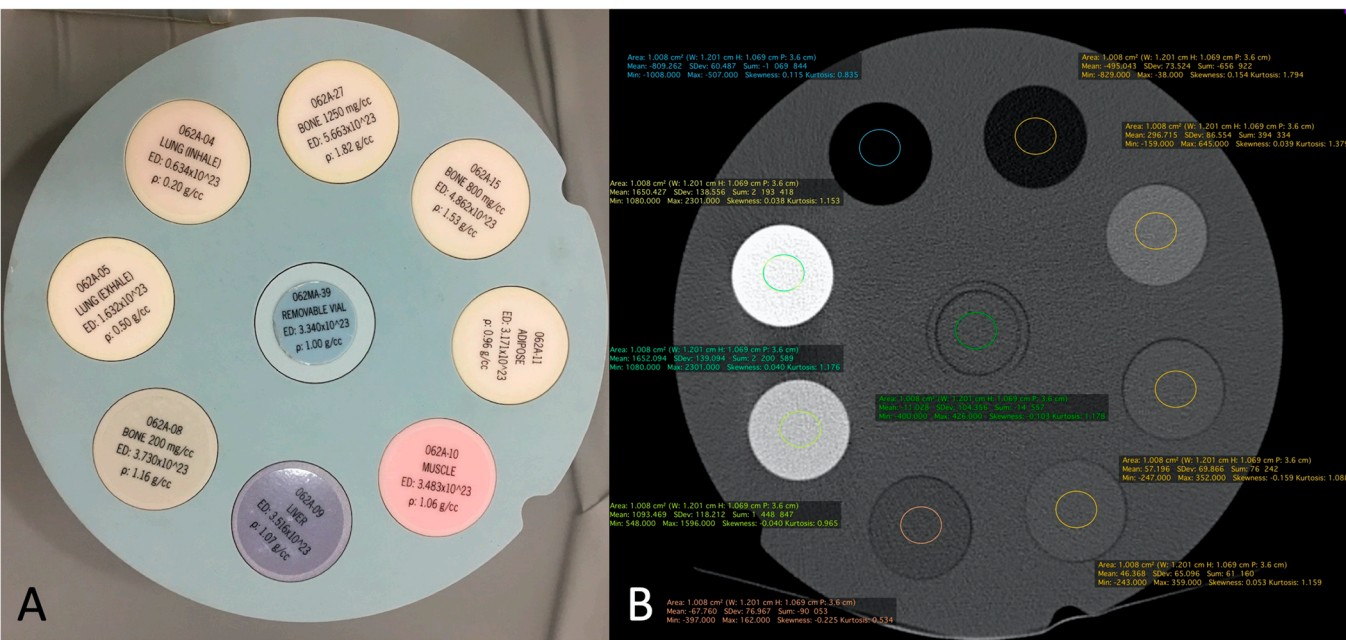

**Figure 3.** The CIRS-062MA Phantom comprises nine inserts with different materials corresponding to different electronic densities (**A**). It is used to calibrate the system in terms of HU for eight different electronic densities corresponding to eight tissues and a standard removable vial for CBCT and MDCT. Example of ROI measurement of each density tunnel (**B**). HU: Hounsfield unit; CBCT: cone-beam computed tomography; MDCT: multi-detector computed tomography; ROI: region of interest.

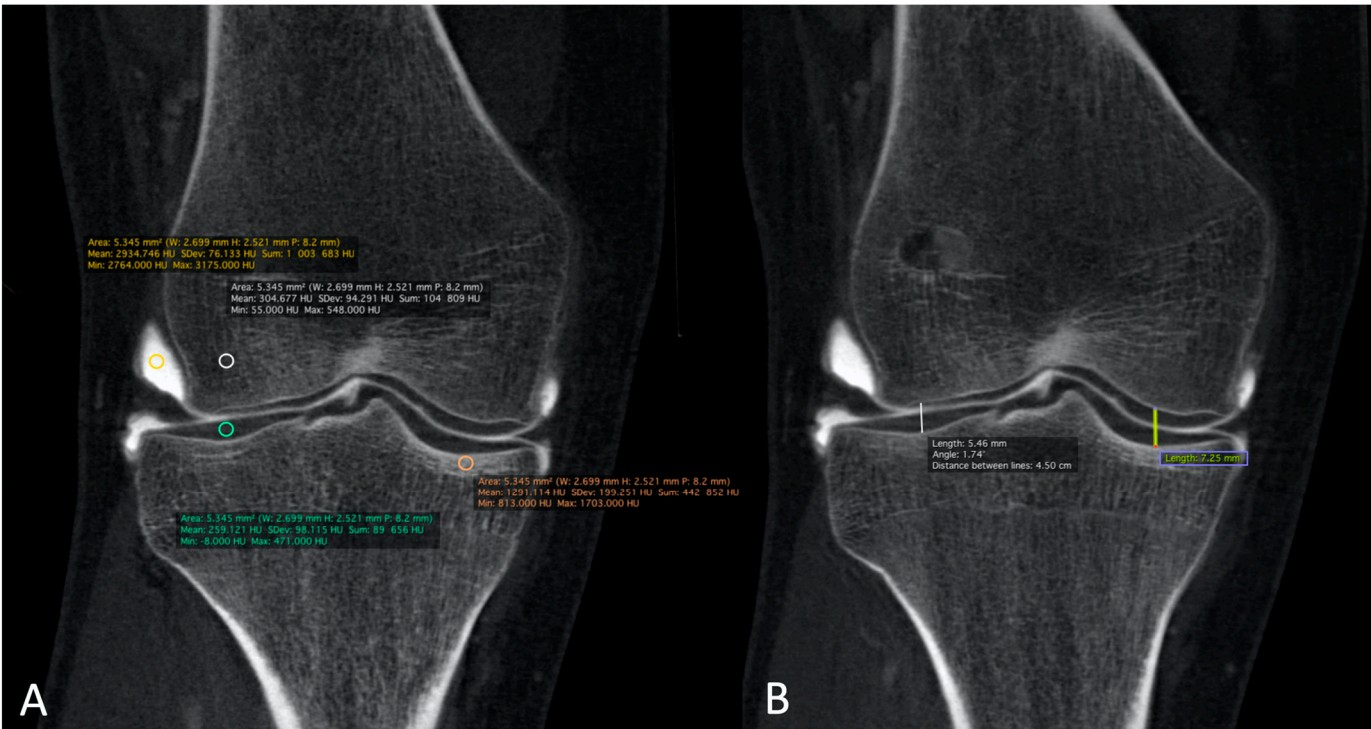

**Figure 4.** The HU numbers of all 32 CBCT-As were analyzed in the coronal plane after the application of MBIR2. Here, an example of a right knee is shown. The HUs for CM, trabecular and cortical sub chondral bone, and for cartilage densities were reported. ROIs were also placed with three measurements for every item (**A**). The MJW was evaluated with the MJW at the lateral and medial side of the femoro-tibial joint (**B**). HU: Hounsfield unit; CBCT-A: cone-beam computed tomography arthrography; ROI: region of interest; MJW: mean joint width; CBCT: cone-beam computed tomography; MDCT: multi-detector computed tomography.

Finally, MJWs were calculated, of the lateral and medial sides for the knee and center of the radio-carpal joint in the wrist or talocrural joint in the ankle, the tibio-talar joint in the ankle, and the femoro-tibial joint in the knee (Figure 4B), and compared with the corresponding DR images.

### 2.5. Radiation Dose Measurement of DR and CBCT-A

To evaluate the radiation exposure for the CBCT-A compared to the DR, a polymethyl-methacrylate (PMMA) phantom with dimensions of $20 \times 15 \times 4$ cm$^3$ (Figure 5) was used to simulate the hand and wrist. Doses to the hand and wrist were estimated by measuring the absorbed dose using a 10 cm cylindrical ionization chamber placed at the center of the PMMA phantom. The absorbed dose was measured for the two scouts (PA, LAT) + one 3D acquisition of the CBCT-A and for the two views (AP + LAT) of the conventional DR.

No effective dose information could be computed based on our measurements. Nevertheless, the measurement of absorbed doses at the center of the phantom remains a valid method for comparing the exposure between the two different modalities [17].

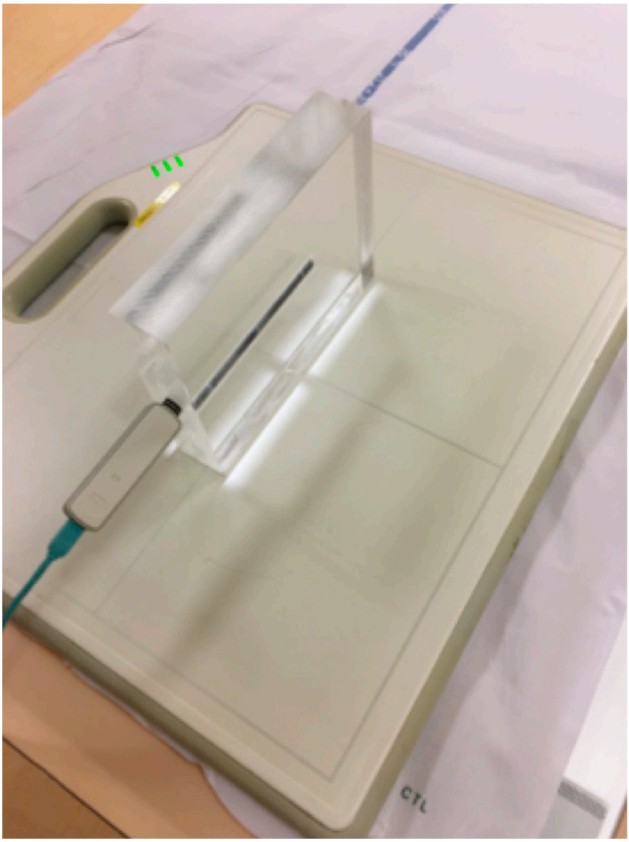

**Figure 5.** Plexiglas phantom (PMMA) used to represent hand or wrist extremities to calculate the absorbed dose of the CBCT and DR imaging modalities, with dimensions of $20 \times 15 \times 4 \text{ cm}^3$ CBCT: cone-beam computed tomography; DR: digital radiography.

*2.6. Statistical Analysis*

First, for the qualitative analysis and to compare the image quality of CBCT-A using the two readers, we used the Kappa coefficient. The interpretation of agreement statistic scores between the two readers for DR and CBCT-A concerning the erosions, sclerosis, MJW, and KLC was carried out using criteria developed by Landis and Koch. Values of 0–0.20 represent slight agreement, 0.21–0.40 fair agreement, 0.41–0.60 moderate agreement, 0.61–0.80 substantial agreement, and 0.81–1 is considered almost perfect agreement [18]. Furthermore, we compared DR with CBCT-A concerning the qualitative assessments of erosions, sclerosis, and quantitative measurements of MJW, as well as KLC for DR and A- CBCT-A using Prism software (Graphic Pad Prism Version 6.0e). Column analyses using *t*-tests were performed. Owing to the non-Gaussian distribution, the Wilcoxon matched pairs signed rank test was employed to compare the results. A *p*-value < 0.05 was considered statistically significant. The Spearman test was calculated for qualitative parameters.

**3. Results**

The qualitative image analysis was excellent in all CBCT-A cases corresponding to the best scale for image quality, with an excellent inter-observer concordance (kappa = 1), as shown in Figure 4, Figure 6, and Figure 7. Twenty-four patients had an OA diagnosis (KLC ≥ 2) with the CBCT-A, and twenty-one were subclassed with DR. No statistically significant difference was observed in terms of sclerosis (*p* = 0.29) and erosion (*p* = 0.184) between both modalities. Examples of OA underestimation, with DR compared with CBCT-A, are shown in Figure 6.

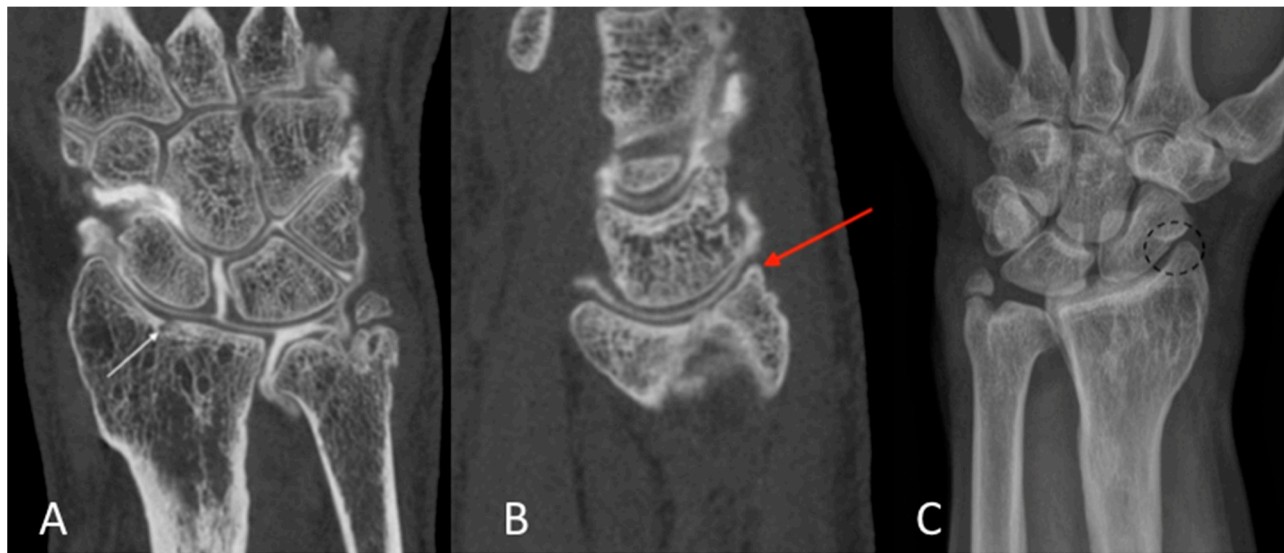

**Figure 6.** CBCT-A of left wrist in coronal (**A**) and sagittal (**B**) planes, which showed sub chondral erosion ((**A**) white arrow) and definite anterior osteophyte of the radial lip ((**B**) red arrow), grade 3 of the KLC scoring system. The AP DR showed a possible joint space narrowing between the scaphoid and radial styloid ((**C**) dark dotted circle), grade 2 of KLC scoring system. The KLC scoring based on DR is underestimated compared to CBCT-A. As osteoarthritis was confirmed between the radius and scaphoid bone surfaces, scaphoidectomy was performed for this patient instead of styloidectomy considered initially based on DR analysis. CBCT-A: cone-beam computed tomography arthrography; DR: digital radiography; KLC: Kellgren and Lawrence classification.

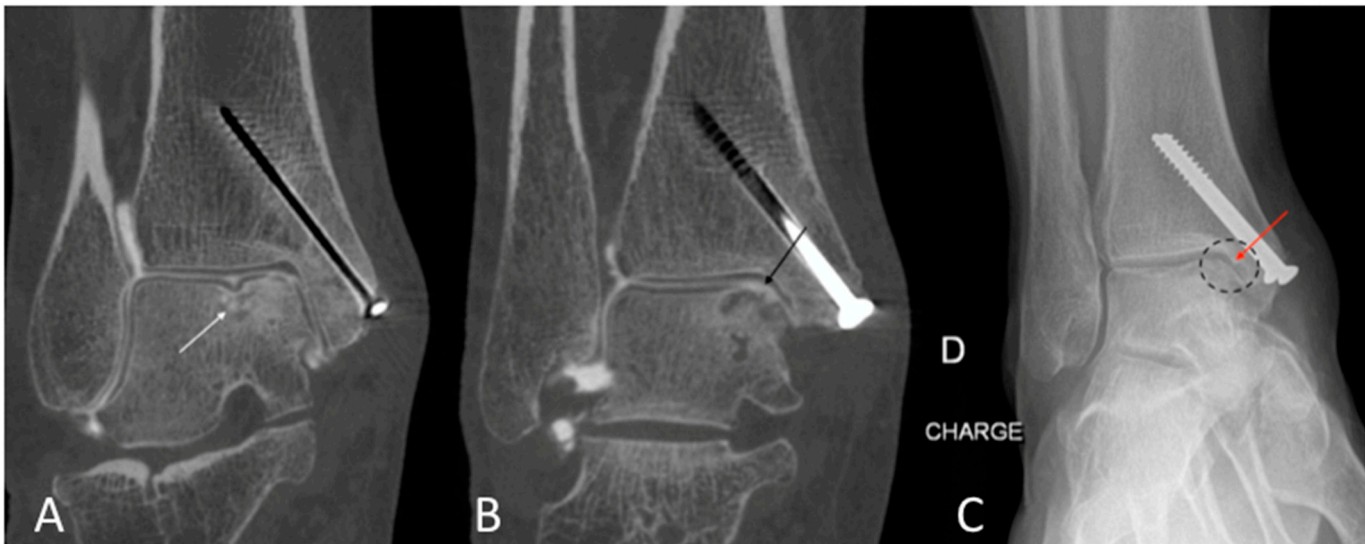

**Figure 7.** CBCT-A of right ankle in coronal (**A**,**B**) planes, showing sub chondral erosion ((**A**) white arrow) and definite medial talo-crural with cartilage loss ((**B**) dark arrow), grade 3 of KLC scoring system. The AP DR showed a possible joint space narrowing ((**C**) dark dotted circle) and a small osteophyte ((**C**) red arrow), grade 2 of KLC scoring system. The KLC grading based on DR under estimated compared to CBCT-A. The presence of material was a limitation to performing MRI in this case. CBCT-A: cone-beam computed tomography arthrography; DR: digital radiography; KLC: Kellgren and Lawrence classification; MRI: Magnetic resonance imaging.

Concerning the quantitative image analysis, OA under-classification was noticed with DR regarding the MJW ($p = 0.02$), detection of osteophytes (<0.0001), and KLC ($p < 0.0001$), as shown in Table 3.

**Table 3.** *p*-values of CBCT criteria in OA grading compared with DR.

|  | **Median** | **Rs (Spearman)** | **p Values** |
|---|---|---|---|
| Mean joint width (MJW) | 0.09876 | 0.9405 | 0.0213 |
| Kellegren and Lawrence | −1.000 | 0.6350 | <0.0001 |
| Osteophytes | 0.0 | 0.4848 | <0.0001 |
| Subchondral sclerosis | 0.0 | 0.3551 | 0.2972 |
| Erosions | 0.0 | 0.3393 | 0.1849 |

CBCT: cone-beam computed tomography; OA: osteoarthritis; DR: digital radiography.

Table 4 summarizes the results obtained from the calibration of the HU with the standardized phantom. Measurements in both the MDCT (Siemens and GE) revealed similar densities in all tissues analyzed. For CBCT, the HU did not correspond to the usual values obtained for the MDCT, with a greater mean deviation obtained with the CT HU for the MBIR1 than for MBIR2. The mean HU calculated for the CBCT-A was 1966 for CM, 328 for trabecular bone, 812 for the subchondral cortex, and 330 for cartilage. We focused on bone density, as no normal values were found in the literature for cartilage; regarding CM, the contrast was diluted with the joint effusion in many cases. We noticed that the HU of CBCT-A measured for trabecular bone and cartilage were similar and corresponded to the bone density values of 200 mg/cc for the MDCT, with either MBIR1 or MBIR2 (236-298HU). The HU of subchondral cortical bone based on CBCT-A was similar to the HU of bone with a density of 80 mg/cc based on MDCT, particularly with MBIR2 (Table 4).

**Table 4.** Mean density values in HU from MDCT (Siemens and GE) and CBCT with MBIR1 and MBIR2.

|  | **GE CT Densities** | **Siemens MDCT Densities** | **CBCT with MBIR 1 Densities** | **CBCT with MBIR2 Densities** |
|---|---|---|---|---|
| Bone of 1250 mg/cc | 1651.115 | 1652.094 | 1895.378 | 1061 |
| (ρ: 1.82 g/cc) | (SD: 149.550) | (SD: 139.094) | (SD: 142.514) | (SD: 97.320) |
| Bone of 800 mg/cc | 1091.529 | 1093.469 | 1282.889 | 858.058 |
| (ρ: 1.53 g/cc) | (SD: 93.369) | (SD: 118.212) | (SD: 118.033) | (SD: 85.509) |
| Bone of 200 mg/cc | 298.070 | 296.715 | 298.396 | 236.401 |
| (ρ: 1.16 g/cc) | (SD: 78.129) | (SD: 73.) | (SD: 78.129) | (SD: 59.354) |
| Adipose tissue | −68.434 | −67.760 | −73.072 | −70.934 |
| (ρ: 0.96 g/cc) | (SD: 77.762) | (SD: 76.967) | (SD: 71.203) | (SD: 62.959) |
| Muscle (ρ: 1.06 g/cc) | 43.208 | 46.368 | 33.949 | 15.139 |
|  | (SD: 68.922) | (SD: 65.096) | (SD: 74.198) | (SD: 57.353) |
| Liver (ρ: 1.07 g/cc) | 56.396 | 57.196 | 42.652 | 26.372 |
|  | (SD: 76.148) | (SD: 69.866) | (SD: 75.826) | (SD: 55.516) |
| Lung (inhale) | −810.597 | −809.262 | −862.192 | −760.330 |
| (ρ: 0.20 g/cc) | (SD: 67.894) | (SD: 60.487) | (SD: 52.165) | (SD: 59.773) |
| Lung (exhale) | −493.688 | −495.043 | −560.997 | −494–122 |
| (ρ: 0.50 g/cc) | (SD: 79.298) | (SD: 73.524) | (SD: 57.972) | (SD: 56.538) |

MDCT: multi-detector computed tomography; CBCT: cone-beam computed tomography; s: second; SD: standard deviation; MBIR: model-based iterative reconstruction.

The absorbed dose to the hand-wrist for the CBCT-A was estimated to be higher than that of the DR; results for the absorbed doses have been presented in Table 5.

**Table 5.** Absorbed dose measured with the PMMA phantom for CBCT-A and DR.

|  | **CBCT-A** | **DR** |
|---|---|---|
| Dose radiation for PA projection mGy | 0.037 mGy | 0.029 mGy |
| Dose radiation for oblique/lateral projection | 0.013 mGy | 0.033 mGy |
| 1 3D acquisition | 4.902 mGy | -- |
| Total dose radiation | 4.952 mGy | 0.062 mGy |

CBCT-A: cone-beam computed tomography; DR: digital radiography; s: second; mGy: milli-Gray.

## 4. Discussion

In our study, we have highlighted that the image quality was excellent in all cases of the CBCT-A study. These results are consistent with those from previous studies [13,20–23], with excellent visualization of the bone microarchitecture, primarily due to the high spatial resolution of this technique [2,9,15,16,23]. To the best of our knowledge, this study is the first to demonstrate the advantages of using the CBCT-A scan for the grading of peripheric joints with OA, as it permits a better treatment. Even though the DR remains the imaging modality of choice in the initial investigation of OA [2–5,7], CBCT-A offers many advantages in investigating OA. It provides superior diagnostic performance and staging for cartilage lesions, as shown in the study of Posadzy et al. [23], with better KLC scoring of OA despite its invasiveness. Furthermore, it allows for a better visualization of cartilage, as this structure is non-radiopaque, and of other intrinsic joint structures, especially in the knee joint (e.g., the meniscus). Penetration of the CM within deeper layers of the cartilage surface indicates an articular-sided defect of the chondral surface [4]. In the study by Carrino et al. [13], the authors were limited by the presence of artifacts, for which no artifact correction algorithm was applied in the first CBCT generation. These artifacts were clearly diminished in our recent device, mostly due to the new iterative reconstruction algorithm installed with MBIR2 rather than with MBIR1. In our study, we also observed that there was an OA under-classification when using DR regarding the MJW, detection of osteophytes, and KLC.

CBCT is an emerged technique with high cost and dose effectiveness in various pathologies. It was found to be superior to MRI, for example for the temporo-mandibular joint, compared to MRI, and this technique even remains mandatory for soft tissue diseases [12]. CBCT has advanced as a valuable novel technique for assessments of osteoarthritis [2]. Finally, the study of Posadzy et al. comparing MRI and CBCT-A highlighted the superiority of CBCT-A for talus cartilage injuries comparing it to MRI [23].

Other clinical advantages of this device were revealed as well. As mentioned in other studies, CBCT allows for the WB investigation of extremities [2,13,15,16,23–30], which still needs to be further studied. Furthermore, the three-dimensional data of CBCT facilitate more quantitative analyses, such as segmentation, erosion detection, characterization of the subchondral bone architecture, and measurements of bone mineral density [13,30].

Regarding a comparison with MRI, this technique is commonly used for osteoarthritis assessments; however, comparing MRI, for example for knee osteoarthritis taking arthroscopy as a gold standard, showed that DR could be sufficient for patients who are not symptomatic [27]. Another study compared CT and MRI and found a strong correlation between the two techniques, and the authors highlight the excellent bone-to-soft tissue contrast based on CT analyses [28]. Finally, as DR-based techniques are essential for bone-related pathologies, and novel techniques, such as CT, like those generated from MRI, are recently compared to CT, and the results demonstrated that similar scores are shown for knee osteoarthritis grading [29].

For CBCT, the HU did not correspond to the usual values obtained for the MDCT, with a mean deviation from the CT HU larger with the MBIR1 at −72% (SSD 221) than the MBIR2 at −52% (SSD 94). The current study is the first to compare these HU values between the MDCT and CBCT with two different iterative reconstruction algorithms: MBIR1 and MBIR2. The better results found with the MBIR2 were probably due to the device progress in the iterative reconstruction algorithm. Several studies have concluded that CBCT cannot be used for quantification purposes due to the poor calibration in the HU [18]. Swennen et al. stated that with CBCT, the "HU" of an organ's voxel depends on the position in the image volume [31]. This means that DR attenuation of CBCT acquisition systems changes according to the position, producing different HU values for similar bony or soft tissue structures in different areas of the scanned volume. The partial gantry rotation causes an asymmetric dose distribution in the anatomical region of interest [21,32,33]. As seen in this study, the HUs were getting closer to the usual values obtained for the MDCT with MBIR2 as compared to MBIR1 (mean deviation from the CT HU of −72% for MBIR1 and −52% for

MBIR2). Thus, we can expect that further improvements in cone-beam reconstruction algorithms and post-processing will reduce this drawback, permitting us to obtain HUs closer to those of the MDCT, thereby enabling quantitative measurements of some diseases, such as bone necrosis and osteopenia follow-ups.

Concerning the radiation dose, CBCT delivers an absorbed dose (AD) that is higher than that delivered by one DR examination (4.92 mGy and 0.062 mGy respectively), with the AD of 4.952 mGy obtained for the CBCT in our study comparing well with that obtained for the Planmed Verity CBCT in the Koivisto et al. study [21]. Indeed, the absorbed dose in Koivisto et al.'s study was evaluated at 1.99 mGy using an anthropomorphic RANDO wrist phantom. Doses to the different tissues were measured, and an average AD was computed. Their device installation worked at approximately the same kVp as our device but with 36 mAs per acquisition compared to our device's 105 mAs. Together with the different method of evaluating ADs, the different mAs used for each system could partly explain why the AD in our study was approximately 2.5 times higher than theirs. Concerning the AD for the conventional AP + LAT XR, Koivisto et al.'s study presented a value of 850 μGy compared to that obtained in our study of 62 μGy. Again, the main difference originates from the mAs used in each case, kVp, and difference in the measurement methodology.

This study displays several limitations. First, we included only a small number of patients given that the CBCT-A scan still displays limited indications, and this technique is only requested by specialized orthopedists, mainly consisting of hand and foot-ankle surgeons. Despite our group's heterogeneity, including wrists, ankles, and knees, this preliminary study primarily sought to position CBCT-A as a technique for preoperative OA management. Based on our data, CBCT-A could be a technique enabling several joint analyses.

As MRI is widely used for osteoarthritis assessments, a comparison between CBCT-A and MRI could be relevant in the future to better select patients. In addition, we recognize that CBCT-A is insufficient for the detection of subchondral edema, an important finding in osteoarthritis disease.

The lack of an HU standard based on CBCT systems is another drawback. We, however, believe that improvements with new devices and reconstruction algorithms will allow for a better assessment of bone density and crucial point prosthetic surgery.

The dose comparison among the different modalities can be performed by assessing the measured absorbed doses [24], though authors are aware that it is common and widely accepted to compare exposures from different modalities while using the effective dose. At present, nevertheless, there are not any readily available methods to evaluate the effective dose (ED) from a hand or wrist examination, i.e., conversion factors from absorbed doses to effective doses for DR or CBCT examinations. The evaluation of the ED could be performed by measuring the organ absorbed doses in the wrist area by inserting dosimeters in a custom-made anthropomorphic RANDO wrist phantom, as performed by Koivisto et al. [24–27].

## 5. Conclusions

In conclusion, the CBCT-A is more reliable for OA diagnoses than DR, as revealed in our results using a three-point rating scale for the qualitative image analysis, with higher quality and an acceptable dose. We entirely agree that the dose comparison by means of mGy (CT dose index) provides the impression of significant doses being used based on CBCT. However, we would like to remind the reader that doses using DR techniques prove to be insignificant, with the effective dose in mSv being significantly lower, as the conversion factor for extremities is very low.

Furthermore, the use of this imaging technique permits mainly the preoperative assessment of extremities in OA diagnoses, and the upright position and the bone microarchitecture analysis are two other advantages of CBCT-A.

**Author Contributions:** Conceptualization and methodology: S.B., M.S.M. and M.H.; Formal analysis: S.B., M.S.M., K.G. and A.N.; Investigation: S.B., M.S.M., K.G. and A.N.; Writing—original draft preparation: M.H., S.B. and M.S.M.; Review: M.H., S.B., M.S.M., K.G., A.N. and P.-A.P.; Editing: M.H., S.B. and M.S.M.; Visualization: M.H., S.B. and M.S.M.; Project administration: M.H. and S.B.; All authors have read and agreed to the published version of the manuscript.

**Funding:** This research received no external funding.

**Institutional Review Board Statement:** This prospective study was performed after the Geneva (Switzerland) cantonal ethics committee research (CCER) approval and in accordance with the guidelines of the Helsinki declaration. The number of the CCER-HUG Geneva approval is 2017-01276, date of approval: August 2017.

**Informed Consent Statement:** Informed consent was obtained from all subjects involved in the study.

**Data Availability Statement:** The data presented in this study are available on request from the corresponding author.

**Conflicts of Interest:** The authors declare no conflict of interest.

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
