# Peer review of "The Role of Cone-Beam Computed Tomography CT Extremity Arthrography in the Preoperative Assessment of Osteoarthritis"

_tomography, doi:10.3390/tomography9060167_

Round 1

Reviewer 1 Report

Comments and Suggestions for Authors

Good study of compare CBCT-A to digital radiography in OA.

1. Why this study did not inculde exclusion criteria?

2. Figure 1 need to be reorganized (red underline).

3. Figure 6 legend did not fit the figure content.

4. Please also show case analysis of ankle joint.

Author Response

The authors would like to thank you for the time and effort you dedicated to reviewing our manuscript. We believed it has now been improved.

This new version has included the suggestions and corrections you made as reviewer. Indeed, we have incorporated all requested changes in the annotated revised manuscript version. Please find in the attachment our point-by-point answers to your queries and comments.

Reviewer 2 Report

Comments and Suggestions for Authors

The work seems well written and structured to me.

Only minor changes:

MDCT HU – Not written in full in the abstract

CBCT mentioned in line 65 and defined in line 68

Figure 6 image B and arrows are missing

Author Response

(The authors gave the same response as above.)

Reviewer 3 Report

Comments and Suggestions for Authors

 The aim of this study was to better define the position of CBCT-A in the diagnosis of OA compared to DR and to improve the use of this technique by evaluating the clinical image quality, radiation dose and bone density quantification of CBCT-A in the musculoskeletal domain. The paper showed that CBCT-A is superior in detecting early arthropathic changes. However, I must point out some major problems.

Introduction

1.The authors describe the importance of accurate diagnosis of KL grade in orthopaedic surgery. However, in current practice, the indication of surgical technique based on KL grade alone is rare. The effect of an accurate diagnosis of KL grade on the surgical decision should be specifically demonstrated.

2.Intramedullary lesions are currently the focus of attention in arthropathy, but these are lesions that cannot be detected by CT. If it is to be argued that CBCT-A is useful in the diagnosis of early arthropathy, its benefit over MRI needs to be demonstrated.

Materials and Methods.

3. The 32 cases analyzed in this study were referred to the authors for preoperative imaging assessment. The usefulness of this imaging assessment should be made easier to understand by indicating not only the lesion but also the planned surgical technique. In particular, the authors would like to see more detailed information on patients with wrist joints, where there are more frequent cases. For example, in the case of TFCC injuries, conventional arthrography is sufficient for diagnosis.

4. Please describe the importance of performing invasive arthrography when MRI is available.

5. Figure 3 has no legend.

6. Figure 6 does not have a sagittal image.

Discussion

7. If the authors state that CBCT is useful in the diagnosis of detailed arthropathic changes and early arthropathy, the comparison with MRI should be discussed.

Author Response

(The authors gave the same response as above.)

Reviewer 4 Report

Comments and Suggestions for Authors

The study aimed to evaluate the image quality and dose of cone-beam computed tomography arthrography (CBCT-A) and compare them to digital radiography (DR) for osteoarthritis (OA) diagnosis. The study found CBCT-A to be more reliable for OA diagnosis than DR. The study is significant as this imaging technique permits the preoperative assessment of extremities in OA diagnosis. The paper addresses a relevant clinical problem: the need for improved imaging techniques for OA. Overall, the study has grammatical errors with poor quality of figures.

The study should be further improved based on the following recommendations:

·        Define all acronyms before use. E.g. HU, MBIR1, and MBIR2 in abstract, CBCT in line C, and CM in line 132

·        Lines 66-67: Please add a reference for the information presented.

·        The introduction could benefit from a more detailed explanation of why there's a need for alternative imaging techniques like CBCT in the context of OA

·        Please add the hypothesis of the study

·        Figure 1: Please correct the typos and change “A-CBCT” to “CBCT-A” throughout the manuscript.

·        Table 2 caption: Remove duplicates and define kVp and mGy

·        Figure 3: Add a caption. The text is not legible

·        Figure 4 caption: Remove duplicates. The text is not legible

·        The study lacks detailed statistical analysis. It would be beneficial to include confidence intervals and other statistical measures to support the findings.

·        Figure 6: Need to add figures for A and B. It would be helpful to include more visual aids, such as side-by-side images from DR and CBCT-A, to visually compare the quality and details captured by each modality.

·        The use of a three-point rating scale for qualitative image analysis is mentioned, but the actual results are not provided.

·        A more in-depth discussion on the clinical implications of the findings would be beneficial. For instance, how would the use of CBCT-A impact patient management, treatment decisions, and outcomes?

·        References: Please include a more recent literature for comprehensive analysis

Comments on the Quality of English Language

Please proof read the manuscript and rectify the grammatical errors.

Author Response

(The authors gave the same response as above.)

Reviewer 5 Report

Comments and Suggestions for Authors

Dear Editor

Thank you for the opportunity to review the paper.

The study's main findings comparing CBCT-A and digital radiography for OA diagnosis were that CBCT-A was more reliable for OA diagnosis than DR, with higher quality and an acceptable dose. The qualitative image analysis was excellent in all CBCT-A cases, with excellent inter-observer concordance, and no statistically significant difference was observed in sclerosis and erosion between both modalities. However, OA under-classification was noticed with DR for MJW, osteophyte detection, and KLC.

The authors should state why they compare the CT with DR not DEXA. DEXA is the best imaging modality for OA diagnosis.

-      How does CBCT-A compare to other imaging techniques, such as MRI, for preoperative assessment of OA?

-      What are the potential benefits and drawbacks of using CBCT-A for OA diagnosis and preoperative planning?

-      The radiation dose should be compared with DEXA as it is obvious DR results in lower radiation dose compared to CT. Also, the Dose could be compared with CBCT for mandible and OPG.

-      The following paper ae useful for radiation dose and discussion https://doi.org/10.3390/reports6040048 , https://doi.org/10.3390/tomography9050141 https://doi.org/10.3390/tomography8060247 

-      What is the study recommendation and limitations 

Author Response

(The authors gave the same response as above.)

Round 2

Reviewer 3 Report

Comments and Suggestions for Authors

I recognized that the authors were generally able to respond adequately to my initial review. However, I consider that a minor revision is required before publication.

I acknowledge that CBCT may be more effective than MRI when metal prostheses are inserted near the joint, as in the case presented by the authors. In addition, I understand from the response to comment #7 that CBCT is reported to be more effective than MRI in cases such as ankle joints. However, MRI is clinically important not only for ligamentous injuries but also for the detection of subchondral bone marrow lesions (e.g. bone marrow edema) in the diagnosis of early osteoarthritis. However, the authors should mention in the limitation that CBCT is inferior to MRI for bone marrow lesions in osteoarthritis, such as bone marrow edema.

Author Response

The authors would like to thank you once more for the time and effort you dedicated to reviewing our manuscript. We believed it has now been improved.

This new version has included your suggestion. Please find in the attachment our answer to your query.

Reviewer 4 Report

Comments and Suggestions for Authors

Thank you for diligently addressing all the comments and making appropriate changes to the manuscript. I just have a few minor recommendations: 

1.     Line 36: Please put the cited reference in “[]” instead of “{}”

2.     Line 73: Please remove “rev 4 com 3”

3.     Figure 1: Please correct the typo of “generation” and change “CBCT-a” to “CBCT-A” under Inclusion criteria

4.     Table 2 caption: DR is defined twice. Please remove duplicate

5.     Figure 4 caption: HU and ROI are defined twice. Please remove duplicates

Author Response

The authors would like to thank you once more for the time and effort you dedicated to reviewing our manuscript. 

This new version has included the corrections you made as reviewer. Indeed, we have incorporated all requested changes in the annotated revised manuscript version. Please find in the attachment our point-by-point answers to your queries.
